# Planar polarization of endogenous ADIP during *Xenopus* neurulation

**Satheeja Santhi Velayudhan[1], Keiji Itoh[1], Chih-Wen Chu[1], Dominique Alfandari[2] and Sergei Y. Sokol[1],***

**ABSTRACT**

Coordinated cell polarity and force-responsive protein localization are essential for tissue morphogenesis, yet how embryonic cells sense forces and respond to mechanical cues remains a challenging question. Afadin- and α-actinin-binding protein (ADIP) has been implicated in microtubule minus-end anchoring, centrosome maturation and ciliogenesis. ADIP is also proposed to associate with the actomyosin cortex and regulate collective cell migration. ADIP behaves as a mechanosensitive planar cell polarity (PCP) protein when overexpressed in *Xenopus* embryos, but the distribution and regulation of endogenous ADIP has been unknown. Here we show that ADIP is present in early ectoderm as randomly distributed puncta that rapidly reorganize and polarize during epithelial wound repair. Endogenous ADIP also becomes enriched and planar polarized in the anterior neural plate towards the midline, consistent with its regulation by mechanical forces that operate during neural tube closure. ADIP polarization is attenuated by depletion of the core PCP component Diversin/Ankrd6, in agreement with the proposed interaction between the two proteins during PCP establishment. Finally, pharmacological disruption of microtubules, F-actin, and nonmuscle myosin II eliminates ADIP polarization in the neuroectoderm, indicating roles for microtubules and actomyosin networks in PCP. Together, these findings suggest that endogenous ADIP senses mechanical cues via the cytoskeletal machinery and functions in a context-dependent manner to control collective cell behaviors during vertebrate morphogenesis.

**KEY WORDS: Afadin- and alpha-actinin-binding protein, SSX2IP, Msd1, Planar cell polarity, Mechanosensitivity, Actomyosin cytoskeleton, Microtubules, Neural tube closure, Nonmuscle myosin II, *Xenopus***

**INTRODUCTION**

Tissue morphogenesis during embryonic development requires precise coordination of cell shape, polarity, and movement across epithelial sheets. One of the key strategies employed during processes such as gastrulation, neurulation, or wound repair is collective cell migration, in which epithelial cells move as a cohesive group while maintaining cell–cell junctions and

dynamically remodeling their cytoskeleton (Davidson and Keller, 1999; Nakamura and Parkhurst, 2024; Rothenberg et al., 2023). These collective movements enable supracellular behaviors such as directed extension, sheet folding, and epithelial tube closure (Angulo-Urarte et al., 2020; Khalil and Friedl, 2010; Lecuit and Yap, 2015). Importantly, the directionality of morphogenetic events is regulated not only by biochemical signals but also by mechanical cues within the tissue, such as tension generated by apical constriction or resistance from surrounding cells (Chu et al., 2025; Swaminathan and Gloerich, 2021). Integration of these physical and biochemical signals is critical for the robustness of developmental programs.

Planar cell polarity (PCP) is a central mechanism for aligning cellular behaviors within the plane of epithelial tissues and is essential for morphogenesis. PCP signaling organizes cells orthogonally to their apical-basal axis and orchestrates processes such as convergent extension, apical constriction, cilia alignment, and neural tube closure (Butler and Wallingford, 2017; Gray et al., 2011; Sokol, 2015). Core PCP components include Frizzled, Dishevelled, Flamingo, Prickle, Diego/Diversin and Van Gogh that localize asymmetrically at cell junctions and relay polarity cues between neighboring cells (Chu and Sokol, 2016; Ciruna et al., 2006; Ossipova et al., 2015b; Yin et al., 2008). In vertebrates, these complexes coordinate actomyosin-dependent processes, such as apical constriction and cell intercalation, to shape and elongate tissues (Devenport and Fuchs, 2008; Matsuda et al., 2023; Newman-Smith et al., 2015; Ossipova et al., 2015a,b,c). Although the upstream signals that establish PCP across epithelial tissues remain poorly understood, emerging evidence suggests that mechanical forces, such as anisotropic tissue tension, can serve as instructive global cues that bias PCP protein localization and link biomechanics to molecular polarity (Chu et al., 2025; Matsuda and Sokol, 2025; Weng et al., 2025).

Afadin- and α-actinin-binding protein (ADIP, also known as SSX2IP or Msd1) is a cytoplasmic protein that was implicated in microtubule minus-end anchoring, centrosome maturation, mitotic spindle orientation and ciliogenesis (Hori et al., 2014; Hori et al., 2015; Klinger et al., 2014). Additionally, ADIP interacts with actomyosin networks and is proposed to function in cell junction remodeling, neural tube closure and wound healing (Asada et al., 2003; Chu et al., 2025; Reis et al., 2021). These studies show that ADIP likely acts in a context-dependent manner, reflecting its subcellular localization in different tissues.

In *Xenopus* embryos, overexpressed ADIP becomes asymmetrically localized in the cells subjected to pulling forces from neighboring cells (Chu et al., 2025). ADIP interacts with the PCP proteins Dishevelled and Diversin, suggesting that it may act as a mechanosensitive regulator of PCP (Chu et al., 2025; Velayudhan et al., 2025). These findings indicate that ADIP coordinates mechanical inputs with PCP signaling during tissue deformation. However, prior studies primarily rely on ADIP overexpression, leaving open the question of whether

[1]Department of Stem Cell Biology and Regenerative Medicine, Icahn School of Medicine at Mount Sinai, New York, 10029 USA. [2]Department of Veterinary and Animal Sciences, University of Massachusetts Amherst, Amherst, MA 01003, USA.

*Author for correspondence (sergei.sokol@mssm.edu)

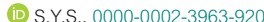 S.Y.S., 0000-0002-3963-9202

endogenous ADIP exhibits similar mechanosensitive properties and polarized localization. To properly evaluate how ADIP functions in mechanotransduction and PCP signaling, the knowledge of endogenous protein distribution is required.

In this study, we raised anti-ADIP antibodies and investigated the spatial distribution and mechanosensitive behavior of endogenous ADIP in the developing *Xenopus* ectoderm and the neural plate. Using quantitative imaging and targeted knockdowns, we show that endogenous ADIP displays planar polarization in response to mechanical cues, requires the cytoskeletal network for its localization, and cooperates with PCP proteins to guide morphogenesis. Our findings suggest that the function of ADIP in vertebrate epithelia is to link mechanical tension and PCP signaling to collective cell behaviors during morphogenesis.

## RESULTS

### Endogenous ADIP forms apical puncta in *Xenopus* ectoderm

To study the distribution of ADIP in the *Xenopus* superficial ectoderm, we first characterized our home-made rabbit anti-ADIP antibody. Negative control staining with rabbit IgG showed no significant cytoplasmic or junctional signal (Fig. 1A-A″). By

contrast, staining with the anti-ADIP antibody revealed distinctive cytoplasmic and cortical puncta in the superficial cells (Fig. 1B-C′). The ADIP puncta frequently associated with cell junctions in stage 10 embryos but were predominantly in the cytoplasm at stage 12, indicating highly dynamic localization. This distribution is similar to the localization of overexpressed ADIP in *Xenopus* ectoderm (Chu et al., 2025; Reis et al., 2021), but distinct from the centrosomal punctate staining reported by others (Barenz et al., 2013; Wang et al., 2021).

Embryos injected with control morpholino oligonucleotide (CoMO) displayed similar cytoplasmic and apical puncta as the uninjected controls (Fig. 1D-D″). In contrast, ADIP MO markedly reduced endogenous staining (Fig. 1E-E″), and quantitative fluorescence analysis confirmed significant reduction of ADIP signal intensity (Fig. 1F). These findings demonstrate that punctate ADIP staining is specific and sensitive to MO-mediated depletion. Immunoblotting revealed a 67 kDa protein band in the control lysates that was strongly reduced in the lysates of ADIP MO-injected tissues (Fig. 1G). Together, these experiments validated the anti-ADIP antibody and confirmed its specificity for endogenous ADIP.

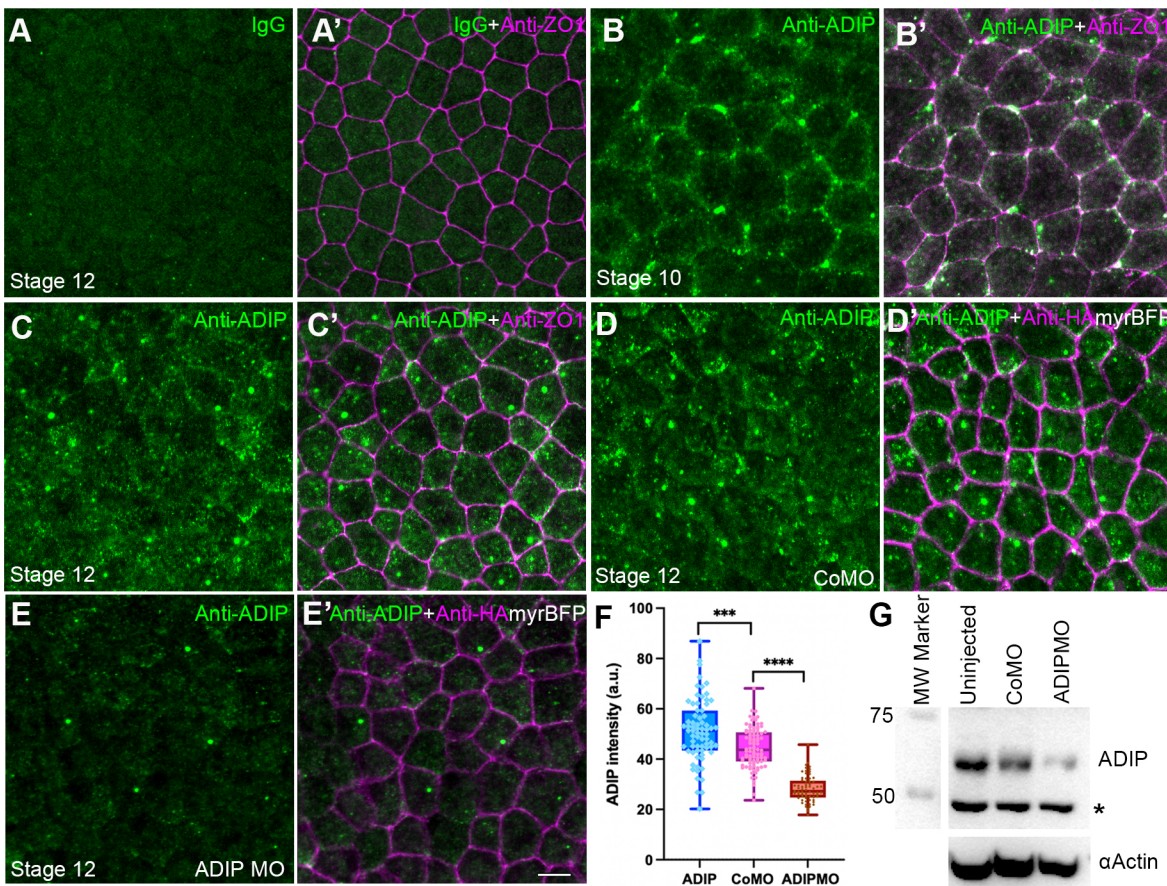

**Fig. 1. Distribution of endogenous ADIP in *Xenopus* ectoderm.** (A-E′) Double staining of stage 12 *Xenopus* ectoderm with anti-ADIP and anti-ZO1 (A-C′) or anti-HA (D-E′) antibodies. (A-A″) Background staining with rabbit IgG (A). ZO1 marks cell borders in the merged image (A′). (B-C′) ADIP is detected predominantly as peri-junctional puncta in stage 10 ectoderm (B) and cytoplasmic puncta in stage 12 (C). (B′,C′) ZO1 marks cell borders. (D-D′) Control MO (20 ng) co-injected with 50 pg HA-myrBFP RNA does not reduce ADIP staining (D), HA-myrBFP labels cell borders in D′. (E-E′) ADIP MO (20 ng) co-injected with 50 pg of HA-myrBFP RNA reduces ADIP puncta (E), HA-myrBFP marks cell borders in E′. Scale bar: 20 μm. (F) Quantification of ADIP fluorescence intensity in uninjected embryos and in embryos injected with control MO and ADIP MO and cultured until at stage 13. Data represent mean fluorescence intensities from three embryos. Welch's *t*-test, *P*<0.001. (G) Anti-ADIP antibody detects 67 kDa protein in stage 12 animal cap lysates by immunoblotting. The 67 kDa band is reduced by ADIP MO but not control MO. Non-specific band (asterisk) is unaffected by ADIP MO. Actin is a loading control (lower panel). Experiments were repeated three times, with 10-15 embryos analyzed per experiment.

## Endogenous ADIP polarizes in the superficial ectoderm during wound healing

Mechanosensitivity of overexpressed ADIP and its planar polarization during wound healing (Chu et al., 2025) prompted us to investigate whether endogenous ADIP displays comparable tension-dependent redistribution in the *Xenopus* ectoderm. ADIP puncta were randomly distributed in gastrula ectoderm, with polarity vectors evenly dispersed across all orientations (Fig. 2A-C). This indicates that endogenous ADIP is not planar polarized in unperturbed ectoderm. Upon epithelial wounding, endogenous ADIP rapidly reorganized toward the wound edge. Within 30 min post-wounding, ADIP puncta became strongly enriched near the junctions facing the wound (Fig. 2D,E). Cell polarity vector mapping showed a clear directional bias toward wound site (Fig. 2F). This behavior closely parallels the redistribution of overexpressed ADIP in response to mechanical cues (Chu et al., 2025).

Cell borders were visualized with DA2B11 (see Materials and Methods), an antibody specific for Wtip, an ADIP binding LIM-domain protein that is also tension-sensitive (Reis et al., 2021). Specificity of DA2B11 was confirmed using Wtip MO knockdown. Control embryos showed robust junctional Wtip, whereas Wtip MO strongly reduced the signal (Fig. 2G,H). Immunoblotting confirmed immunofluorescence data (Fig. 2I). Endogenous ADIP puncta partially overlapped with junctional and perijunctional Wtip

(Fig. 2D-D″), in agreement with published biochemical interactions between Wtip and ADIP at epithelial junctions (Reis et al., 2021). Notably, Wtip also polarized toward the wound (Fig. 2D′), consistent with its proposed role as a mechanosensitive adaptor linking the actomyosin network to junctional scaffolds (Chu et al., 2018; Rauskolb et al., 2022).

## Endogenous ADIP is asymmetrically enriched and planar polarized in neuroectoderm

Puncta of overexpressed ADIP revealed mechanosensitive properties by becoming polarized in the plane of superficial epithelial cells during blastopore lip formation and later, during neural plate folding (Chu et al., 2025). We asked whether endogenous ADIP has a similar distribution in *Xenopus* neurula-stage ectodermal tissues. Immunostaining indicated non-random asymmetric localization of ADIP at the apical cortex of ectoderm cells. In the neural plate of stage 14 embryos, abundant ADIP puncta were polarized toward the dorsal midline in a mirror-symmetric fashion (Fig. 3A-H). This polarization likely reflects a response to a mechanical or chemical cue from the midline. In non-neural ectoderm of stage 15-16 neurulae, endogenous ADIP puncta were oriented towards apically constricting areas at the border of the neural plate (Fig. 3I-L), similarly to GFP-tagged ADIP (Chu et al., 2025). These findings support our previous conclusion that ADIP is a mechanosensitive

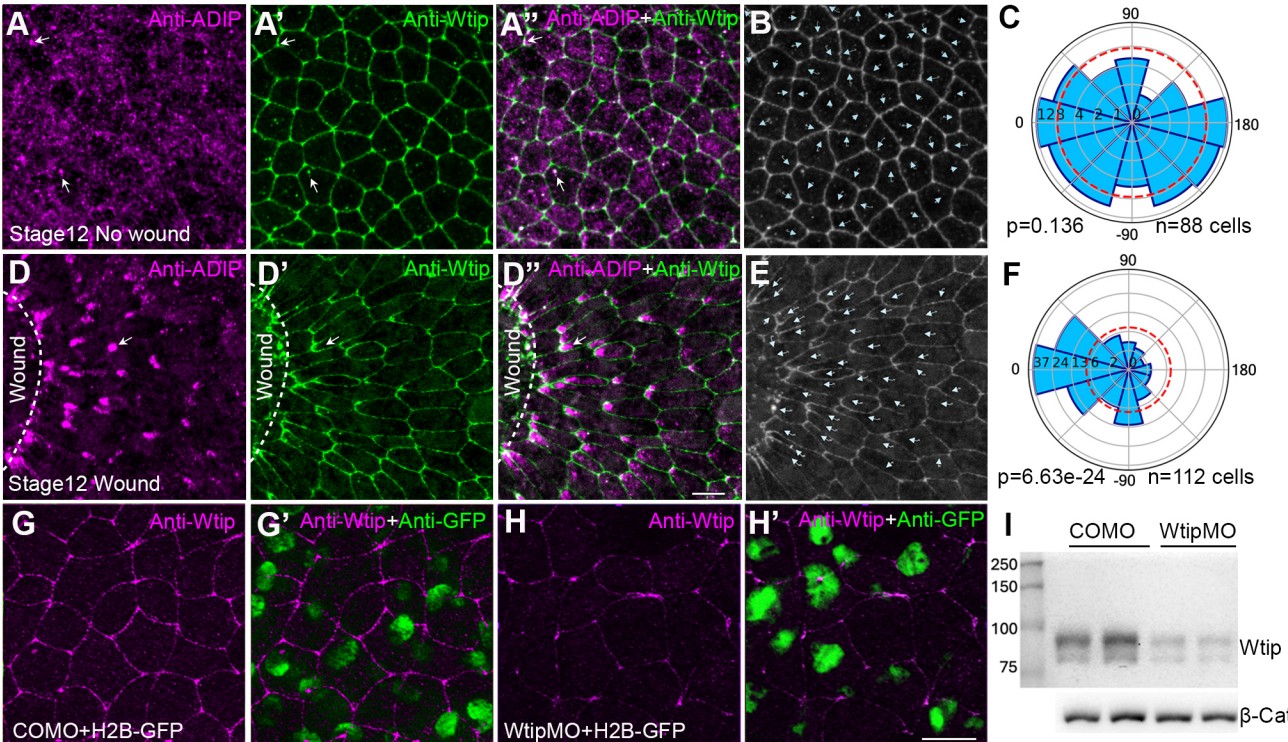

**Fig. 2. Response of endogenous ADIP to wound healing.** (A-C) Immunostaining of stage 12 embryos reveals ADIP puncta in ectoderm cells (A). Cell borders are marked by Wtip (A′); merged image (A″) shows partial colocalization of ADIP and Wtip puncta in the apical domain (white arrows). (B) Segmented cell outlines show random orientation of ADIP polarity vectors (arrows). (C) Rose plot confirms random distribution. (D-F) Stage 12 embryos were wounded on the ventral animal side and allowed to heal for 30 min in 0.7× MMR. ADIP becomes polarized in the cells proximal to the wound margin (D). Wtip labels cell borders (D′); merged image (D″). Arrows in D indicate ADIP puncta and polarized Wtip at cell junction facing the wound. Scale bar in D″: 20 μm; the same scale applies to panels A-A″, B, E, G-G′, H-H′. Cell segmentation (E) and rose plot (F) show ADIP polarity vectors (arrows in E) oriented toward the wound site (0°). Rose plot represents pooled polarity from two embryos. Chi-square test indicates a non-random distribution, $P<0.05$. (G-I) Anti-Wtip staining of stage 10.5 embryos co-injected with 20 ng of control MO (G-G′) or Wtip MO (H-H′) and 50 pg H2B-GFP RNA as lineage tracer. MO-injected cells are marked by rabbit polyclonal antibody for GFP; merged images in G′,H′. (I) Immunoblotting of endogenous Wtip in stage 13 embryo lysates injected with control MO or Wtip MO. Blots were probed with mouse anti-Wtip (DA2B11) antibody (upper panel) and anti-β-catenin antibody as a loading control (lower panel). Data shown are representative of three independent experiments, each performed on 10-15 embryos.

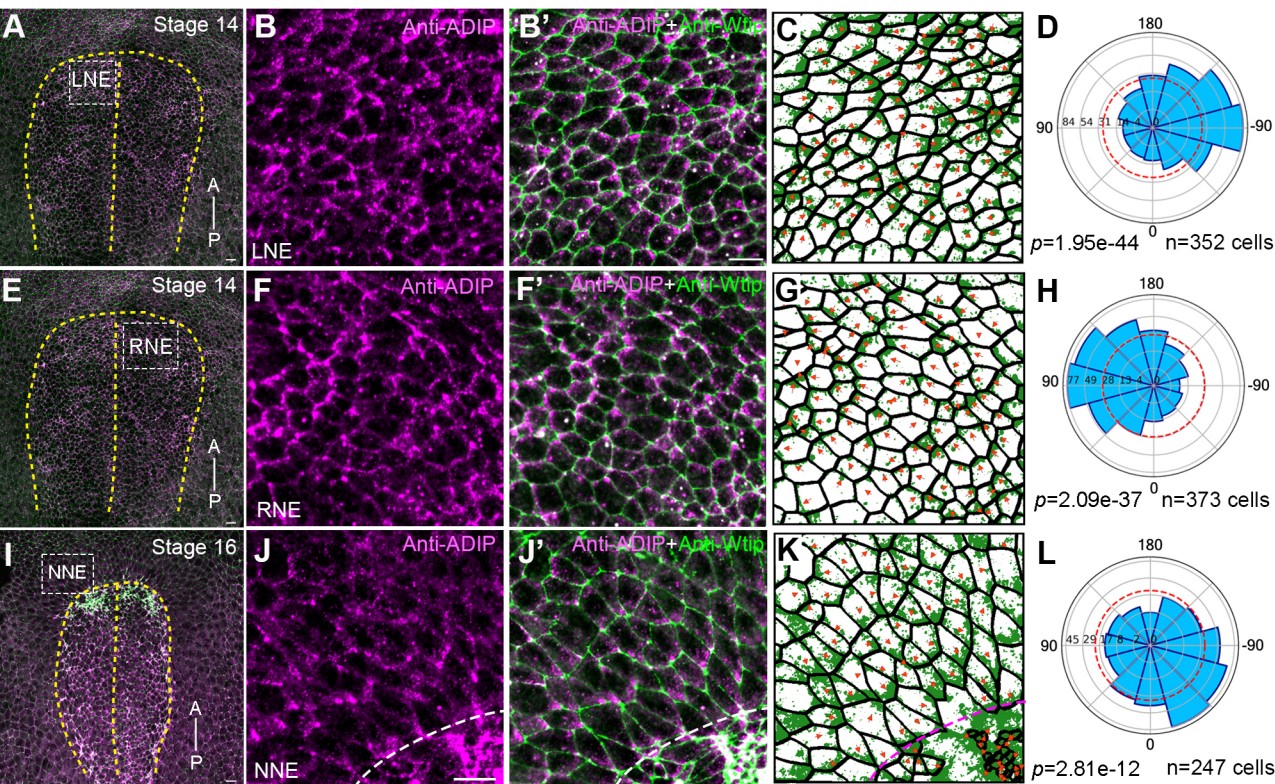

**Fig. 3. ADIP polarization in the neural and epidermal ectoderm of *Xenopus* embryos.** *En face* views of the dorsal surface of representative embryos at stages 14-16 that were double stained with rabbit anti-ADIP and mouse anti-Wtip antibodies. (A,E,I) Low magnification view. Scale bars: 20 µm. (B,B′,F,F′,J,J′) High magnification views of the areas that are boxed in A,E,I, respectively. Scale bars: 20 µm. LNE, left anterior neuroectoderm; RNE, right anterior neuroectoderm; NNE, nonneural ectoderm. Wtip marks cell borders. A-P axis is indicated. Dashed line indicates the dorsal midline in A,E,I and the anterior border of the neural plate in J,K. Following cell segmentation in C,G,K, ADIP puncta enrichment was quantified by rose plots (D,H,L). ADIP puncta are oriented to the right to the midline in A-D, to the left towards the midline in E-H, and to the anterior border of the neural plate in I-L. Polarity vectors were quantified in (C,G,K), and angular distribution is shown in the corresponding rose plot (L). Rose plots represent data from three embryos. High significance of ADIP orientation is confirmed by Chi-square test, *P*<0.001 for all. Each experiment was repeated three times with 10-15 embryos per group.

protein that exhibits planar polarization in response to tensile forces during neurulation.

### Diversin is required for ADIP polarization in neuroectoderm

Diversin is a homologue of the fly PCP protein Diego that facilitates the segregation of core PCP complexes (Feiguin et al., 2001; Jenny et al., 2005). Previous work demonstrated that ADIP and Diversin form a mechanosensitive complex that colocalizes at epithelial junctions and polarizes toward sites of wounding (Chu et al., 2025). Notably, Diversin fails to polarize in ADIP-depleted cells undergoing wound healing (Chu et al., 2025; Velayudhan et al., 2025). We wanted to ask whether the two proteins function interdependently and tested whether Diversin is required for endogenous ADIP polarization in the neural plate. ADIP distribution was assessed in stage 16 control embryos and those depleted of Diversin with previously characterized DivMO (Yasunaga et al., 2011) (Fig. 4A,B).

In embryos injected with a control MO, endogenous ADIP was enriched and medially oriented in the apically constricted cells within the anterior neural ectoderm, forming puncta polarized toward the midline (Fig. 4C,D). Cells injected with ADIP MO exhibited a significant loss of the ADIP signal and appeared apically expanded (Fig. 4E,F). In the majority of DivMO-injected cells, ADIP expression was still detected but its medial enrichment was lost (Fig. 4G-G″). Rose plot analysis confirmed a significant reduction in the polarity vector (Fig. 4H). Quantitative comparison revealed that Diversin depletion caused a significant reduction in the

ADIP signal in neural ectoderm, relative to control MO-injected embryos (Fig. 4I). Together, these findings indicate that Diversin is required for ADIP polarization during morphogenesis.

### Microtubules are necessary for ADIP polarization in the neuroectoderm

Given the dynamic interplay between cytoskeletal elements and polarity signaling and functional association of ADIP with microtubules, we next asked whether microtubules are required for planar polarity of ADIP in the neuroectoderm. Embryos were treated from stage 12 to stage 16 with either 0.1% DMSO or DMSO with 2 µM taxol or 2.5 µM nocodazole in order to stabilize or depolymerize microtubules. The embryos were immunostained for endogenous ADIP and Wtip to mark cell junctions (Fig. 5A,D,G). In DMSO-treated control embryos, endogenous ADIP was asymmetrically enriched in anterior neural ectodermal cells, with clear medial orientation toward the neural plate midline (Fig. 5B,C). Taxol-treated embryos showed a similar pattern, with ADIP puncta maintaining cortical enrichment and planar polarity (Fig. 5E,F), suggesting that stabilized microtubules do not disrupt normal ADIP localization. By contrast, embryos treated with nocodazole exhibited disrupted ADIP polarity. ADIP puncta were no longer asymmetrically distributed and appeared randomly oriented across cells (Fig. 5G,H). Rose plot analysis confirmed that ADIP polarity vectors were significantly randomized following microtubule disruption, indicating a loss of planar polarity.

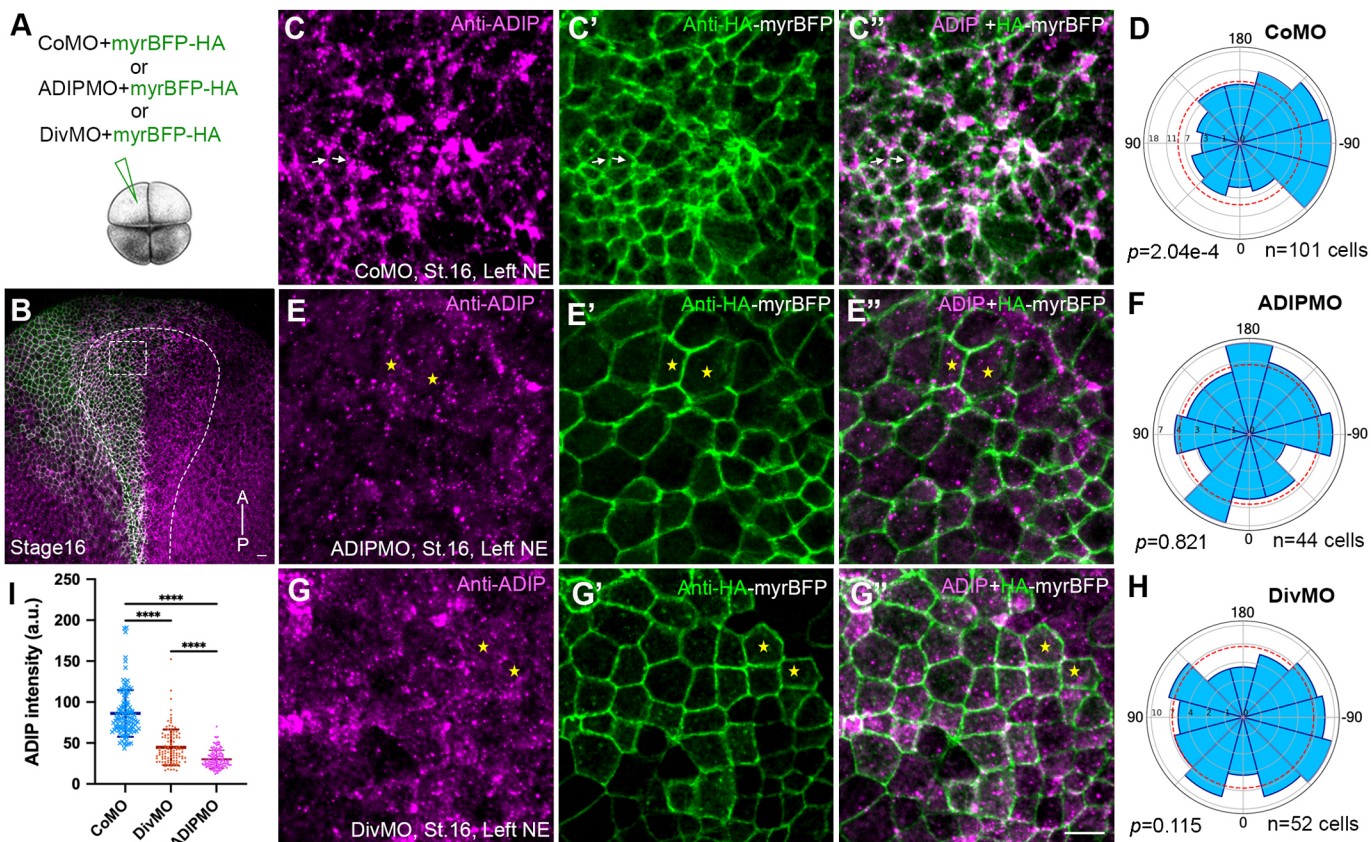

**Fig. 4. Diversin knockdown disrupts ADIP polarization.** (A) Experimental scheme. CoMO, ADIP MO, or Diversin-MO (DivMO), 20 ng each, were injected into the left dorsal blastomere of four-cell *X. laevis* embryos to target neural ectoderm. MyrBFP-HA RNA was co-injected as a membrane lineage tracer. (B) *En face* low magnification view of stage 16 embryo immunostained for ADIP and HA. Scale bar: 20 µm. The anterior-posterior (AP) axis and neural plate boundary are marked; dashed box indicates the region magnified in panels C,E,G. (C-C″) Control MO. Arrows mark clearly polarized cells. (E-E″) ADIP MO reduces ADIP fluorescence (E) and expands apical domain. (G-G″) DivMO. Cells lacking ADIP polarity are marked by asterisks. Scale bar: 25 µm. (D,F,H) Rose plots quantify ADIP polarity. (I) Quantification of ADIP intensity in cells injected with DivMO (*n*=74 cells), and ADIP MO (*n*=58 cells), as compared to control MO (*n*=107 cells). Means and s.d. are shown on graphs; one-way ANOVA, ****$P<0.0001$. Results are representative of three experiments, with more than ten embryos in each group.

These results demonstrate that intact microtubules are necessary for ADIP neuroectodermal polarization and lead us to propose that microtubule-based transport regulates the asymmetric localization of ADIP and/or other PCP components.

### ADIP planar polarity requires actomyosin contractile network

Cellular contractility is controlled by the activity of non-muscle Myosin II (Vicente-Manzanares et al., 2009). Rho-associated kinase (ROCK) promotes actomyosin network assembly by direct phosphorylation of myosin light chain (MLC) and myosin phosphatase MYPT1 (Amano et al., 2010; Kimura et al., 1998). Inhibition of ROCK disrupts epithelial folding and delays neural tube closure in mouse embryos (Escuin et al., 2015). To determine whether cytoskeletal tension is required for ADIP planar polarity, we disrupted components of the RhoA-ROCK-myosin-actin pathway by incubating embryos with established pharmacological inhibitors (Fig. 6A).

Control embryos cultured in 0.1% DMSO, 0.1× MMR exhibited robust planar polarization of endogenous ADIP in the anterior neural ectoderm at stage 16, with puncta oriented towards the midline (Fig. 6C,D). By contrast, treatment with the ROCK inhibitor Y-27632 (50 µM) disrupted ADIP asymmetric accumulation (Fig. 6E,F). Treatment of embryos with 25 µM Blebbistatin, a non-muscle

myosin II inhibitor (Straight et al., 2003), similarly abolished ADIP planar polarity (Fig. 6G,H). Finally, depolymerization of F-actin with 2.5 µM cytochalasin D resulted in embryos lacking detectable ADIP polarization (Fig. 6I,J). Wtip polarization was less apparent in this model.

Thus, disrupting tension-generating systems at multiple levels including ROCK, myosin II, and F-actin significantly impairs ADIP asymmetry. Together, these observations suggest that mechanosensitive planar polarization of ADIP requires both intact microtubules and the actomyosin contractile network.

### DISCUSSION

This study investigated the distribution of endogenous ADIP in both neural and non-neural ectoderm of *Xenopus* embryos. Immunostaining with home-made polyclonal antibodies show that ADIP is distributed as apical puncta in superficial ectoderm cells. Upon wounding, ADIP rapidly redistributes toward the junctions facing the wound, supporting a role of ADIP in mechanotransduction during tissue remodeling. During neurulation, endogenous ADIP displays robust medial enrichment in the anterior neural plate and polarizes toward apically constricting neighboring cells at the neural plate border. These patterns recapitulate the behavior of overexpressed ADIP during tissue folding (Chu et al., 2025), indicating that endogenous ADIP is similarly responsive to mechanical stimuli. In

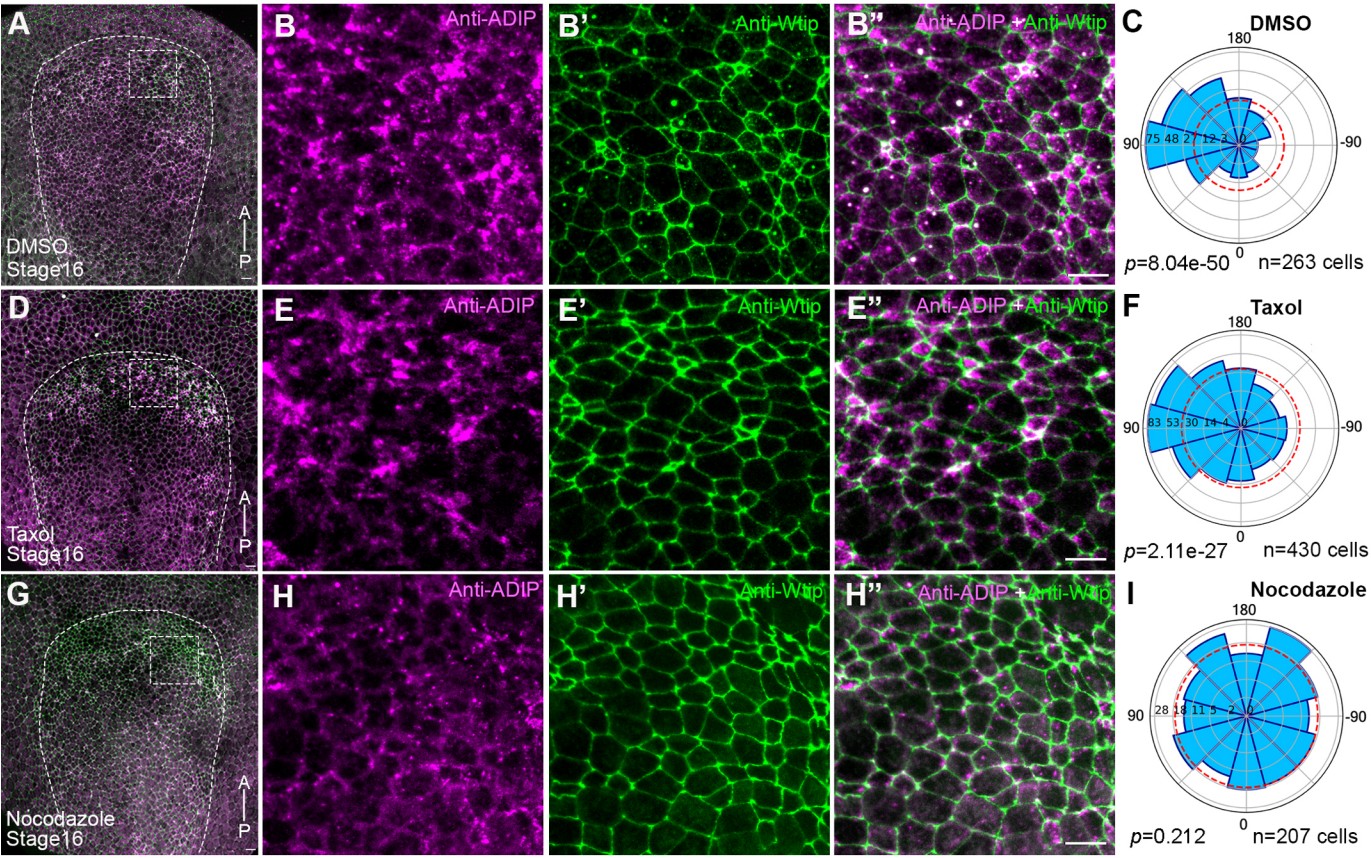

**Fig. 5. Planar polarity of ADIP in neuroectoderm requires microtubules.** (A,D,G) Neural plates of stage 16 embryos were treated with DMSO (negative control, A), 2 µM Taxol (positive control, D), or 2.5 µM nocodazole (G) from stage 12 to stage 16 and immunostained for ADIP and Wtip. The anterior–posterior (AP) axis is labeled, and the neural plate is outlined with dashed lines. Scale bar: 20 µm. (B,E,H) Boxed regions correspond to the anterior neural ectoderm. Wtip marks cell borders in B′-B″,E′-E″,H′-H″. Colocalization of ADIP and Wtip puncta is visible. Scale bar: 20 µm. Fluorescent images represent three independent experiments, each consisting of 15 embryos per group. (C,F,I) Rose plots quantify ADIP polarity vectors after the DMSO (C), taxol (F) and nocodazole (I) treatments, respectively. Data were obtained for three embryos per group. Chi-square test indicates non-random ADIP puncta orientation in C and F, $P<0.0001$.

addition to physical stresses associated with the apical constriction at the border of the neural plate (Chu et al., 2025), the medial orientation of ADIP puncta within neuroectoderm suggests the existence of pulling forces at the dorsal midline. These forces may originate from the apically constricting midline cells (Matsuda et al., 2023) or may be due to the adhesion between neuroectoderm and the developing notochord. These observations indicate that ADIP senses local mechanical environments and becomes asymmetrically enriched along the force axis, consistent with its hypothetical role in guiding collective epithelial cell behaviors. Alternatively, ADIP may be responsive to some biochemical signal produced by the midline.

Our results demonstrate that microtubules are essential for the polarization of ADIP in the neural ectoderm. ADIP has previously been characterized as a microtubule-anchoring protein involved in centrosome maturation and spindle orientation (Barenz et al., 2013; Hori et al., 2014, 2015). Notably, oriented microtubule bundles are required for establishing and maintaining tissue polarity in multi-ciliated epithelia and neurons (Devenport, 2014; Dorrego-Rivas et al., 2022; Vladar et al., 2012). We propose that ADIP acts as a cortical anchor for non-centrosomal microtubule arrays that align with tissue-level tensile forces during neurulation. Supporting this possibility, polarized non-centrosomal microtubule arrays provide directional cues that orient the asymmetric localization of core PCP proteins at *Drosophila* apical junctions (Matis et al., 2014). Similarly, in *C. elegans* embryos, protein complexes originating

from the pericentriolar material can also function as non-centrosomal microtubule-organizing centers (MTOCs) (Sanchez et al., 2021). In addition, apical microtubules can bear compressive forces and are coupled to adherens junctions by PCP signaling to regulate epithelial cell shape (Singh et al., 2018). Together, these observations support a model, in which ADIP integrates microtubule organization and epithelial PCP signaling. Consistent with this hypothesis, cortical microtubules are aligned along the force axis (Chien et al., 2015; Roper, 2020), and ADIP can be delivered via microtubule motors dynein and Kinesin-14 (Barenz et al., 2013; Yukawa et al., 2015).

Besides microtubules, our inhibitor experiments highlight a critical requirement for the actomyosin cytoskeleton in ADIP polarization. Treatment with Blebbistatin, Cytochalasin D, and a ROCK inhibitor disrupted ADIP's asymmetric localization in the neural plate. Currently, it is difficult to distinguish a direct effect of these inhibitors on trafficking from an indirect effect on ADIP via the disruption of mechanical force generation. Nevertheless, the interaction of ADIP with both microtubular and actomyosin networks suggests that ADIP may function in different tissues in a context-dependent manner. In some epithelial cells, the association of ADIP with known mechanosensitive modulators of cell junctions, α-actinin and Afadin (Asada et al., 2003) may form a network sensing the direction of actomyosin contractions and establishing PCP to coordinate force transmission across tissue. In other tissues, such as

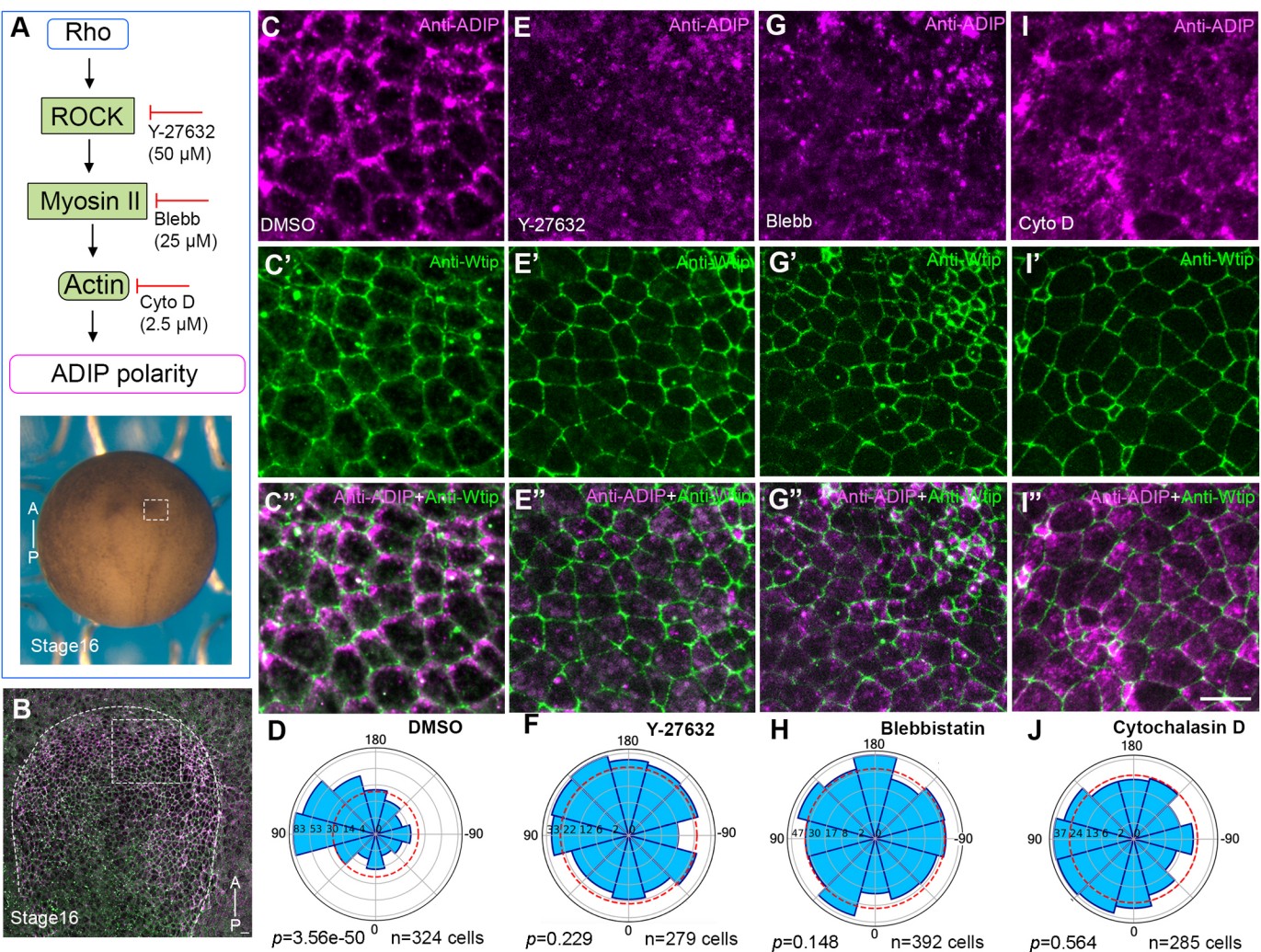

**Fig. 6. Actomyosin contractility and F-actin integrity are necessary for ADIP planar polarization in neuroectoderm.** (A) Experimental scheme. Embryos were treated with different drugs to inhibit the RhoA–ROCK–Myosin pathway and were double stained at stage 16 for ADIP and Wtip. (B) Low magnification dorsal view of stage 16 neurulae. Scale bar: 20 µm. Dashed boxes denote the anterior neural ectoderm shown at higher magnification in C-C″,E-E″,G-G″ and I-I″. The anterior-posterior (AP) axis is labeled, and the neural plate is outlined with dashes. Images (C-I″) represent three independent experiments with 10-15 embryos per group. (C-C″) Control embryo (0.01% DMSO). (D) Rose plot indicates significant planar polarization of ADIP. (E-E″) ROCK inhibitor Y-27632 (50 µM) disrupts ADIP polarity. (F) Rose plot reveals loss of ADIP polarity. (G-G″) Blebbistatin (Blebb, 25 µM) similarly abolishes ADIP polarity. (H) Rose plot confirms loss of polarity. (I-J) Depolymerization of F-actin with 2.5 µM cytochalasin D (Cyto D) diminishes ADIP orientation. ADIP (I) and Wtip (I′) staining, along with merged image (I″), show reduced medial enrichment; scale bar: 20 µm. (J) Rose plot shows loss of polarity.

mucociliary epithelium, ADIP complexes may be involved in the control of rotational planar polarity of the basal bodies (Butler and Wallingford, 2017).

The requirement for actomyosin contractions in ADIP polarization and the critical role of ADIP in morphogenesis (Chu et al., 2025) reinforce the concept that ADIP is not merely a passive outcome of tissue strains but an active transducer of mechanical and biochemical cues. In regions undergoing morphogenesis, such as the neural plate or the wound edge, apically constricting cells produce directional mechanical cues across the epithelial sheet. In response to these cues, ADIP associates with Diversin and Dishevelled (Chu et al., 2025; Velayudhan et al., 2025) to form PCP puncta that mediate cytoskeletal remodeling and propagate the signal further along tissue plane. Consistent with this idea, the knockdown of Diversin disrupted ADIP planar polarity and impaired apical constriction. These results parallel our previous work showing that ADIP is essential for force-responsive localization of PCP proteins during neurulation and wound healing (Chu et al., 2025; Velayudhan et al., 2025). The

interdependence of ADIP and Diversin suggests that they function together as a mechanosensitive PCP module that responds to tissue tension to transform mechanical cues into spatially restricted PCP complexes critical for epithelial morphogenesis. Our study supports the existence of this module as a bona fide cellular entity (provisionally named the 'tensosome'), extending previously observed tension-dependent behavior of overexpressed ADIP complexes (Chu et al., 2025). The detailed analysis of the composition and function of the endogenous mechanosensitive ADIP complexes awaits future studies.

## MATERIALS AND METHODS
### Plasmid constructs, RNA synthesis and morpholinos

pCS2-myr-tagBFP-HA was previously described (Matsuda et al., 2023). pCS-H2B-EGFP was a gift from Sean Megason (Harvard Medical School, Boston, MA, USA) (Addgene plasmid #53744; RRID:Addgene_53744) (Megason, 2009). Capped mRNAs were synthesized from linearized DNA templates using the mMESSAGE mMACHINE SP6 Kit (Thermo Fisher Scientific) and purified with the RNeasy Mini Kit (Qiagen).

MOs were synthesized by GeneTools (Philomath, OR, USA) and had the following sequences: ADIP MO (5′-TAACTCCTCGACTCCTTCTGGA-CAG-3′) (Reis et al., 2021), Diversin MO (5′-GGCCACATCCTGCTGG-CTCATGAAT-3′) (Yasunaga et al., 2011), Wtip MO (5′-TGTCC-TCATCGTACTTCTCCATGTC-3′) (Chu et al., 2016) and control MO (5′-GCTTCAGCTAGTGACACATGCAT-3′).

### *Xenopus* embryo culture, microinjections, wound healing and drug treatment

*Xenopus laevis* were handled in accordance with the Guide for the Care and Use of Laboratory Animals, NIH and with approval from the IACUC of the Icahn School of Medicine at Mount Sinai. *Xenopus laevis* eggs were fertilized *in vitro* and embryos were cultured as previously described (Dollar et al., 2005). Embryos were staged according to Nieuwkoop and Faber (1994). For microinjections, four-cell stage embryos were placed in 3% Ficoll 400 (Sigma-Aldrich) in 0.5× MMR buffer (50 mM NaCl, 1 mM KCl, 1 mM CaCl$_2$, 0.5 mM MgCl$_2$, 2.5 mM HEPES, pH 7.4), and 10 nL of 50 pg myr-tagBFP-HA or H2B-EGFP mRNA was co-injected with either 20 ng of control MO, ADIP-MO, Diversin MO or Wtip MO into one of the dorsal blastomeres. Wounds were generated by creating a small circular lesion in the animal epithelium of devitellinized stage 12 embryos using fine dissection forceps. Embryos were then allowed to heal for 30 min in 0.7× MMR solution. Following wound healing, embryos were fixed in Dent's fixative (80% methanol, 20% DMSO) (Dent et al., 1989) and immunostained for ADIP and Wtip. To assess the effects of cytoskeletal disruption on endogenous ADIP polarization, embryos were treated from stage 12 to stage 16 with 0.1% DMSO in 0.1x MMR culture buffer (control), or the buffer containing 2.5 μM nocodazole, 2 μM taxol, 25 μM blebbistatin (Sigma-Aldrich), or 2.5 μM cytochalasin D (Focus Biomolecules, PA, USA).

### Generation of the ADIP and WTIP antibodies

*Xenopus* ADIP-specific immunogen was produced as glutathione S-transferase (GST) fusion protein by subcloning the DNA encoding ADIP C-terminal amino acid residues 295-554 into pGEX vector (Itoh et al., 2009). The GST-ADIP(295–554) recombinant fusion protein was expressed in *E. coli*, affinity purified by glutathione-agarose beads, and used to immunize rabbits (Cocalico, PA, USA). Antibodies were affinity purified from the immune sera using GST-ADIP(295–554) immobilized on the PVDF membrane. The resulting rabbit polyclonal antibody (ADIP-15RC) was stored in 50% glycerol at −20°C.

To generate monoclonal antibody to Wtip, Flag-tagged full-length *Xenopus* Wtip in pCS2 was expressed in 293T cells, immunoprecipitated by anti-D agarose beads (ABM) and used to immunize Balb/c mice. Splenocytes from immunized mice were fused with the SP-20 myeloma cell line to generate hybridomas, which were screened by immunofluorescence on HEK293T cells expressing RFP-tagged full-length *Xenopus* Wtip. Positive clones, including DA2B11, were further validated by capillary immunoblotting (WES, Bio-Techne) of lysates from control and RFP-Wtip-expressing embryos and embryos injected with Wtip MO, using previously described protocols (Horr et al., 2023). The DA2B11 hybridoma was deposited to DSHB (University of Iowa, RRID:AB_2876379).

### Immunoblotting, immunostaining and embryo imaging

Immunoprecipitation and immunoblotting were performed as previously described (Itoh et al., 2021). Endogenous ADIP was detected by lysing 30 stage 11 embryos in 520 μl of ice-cold lysis buffer [1% Triton X-100, 50 mM NaCl, 1 mM EDTA, 50 mM Tris-HCl, pH 7.6 supplemented with 1 mM phenylmethylsulfonyl fluoride (PMSF), 10 mM NaF, and 1 mM Na$_3$VO$_4$]. Yolk was removed by centrifugation at 13,000 rpm for 5 min at room temperature. Proteins were analyzed after adding an equal volume of 2× sample buffer. One embryo equivalent of total protein (from five embryos) was boiled in the presence of 2% 2-mercaptoethanol, separated by SDS-PAGE, and transferred to Immobilon-P PVDF Membrane (Sigma-Aldrich). Immunoblotting was performed with rabbit anti-ADIP polyclonal antibody or mouse anti-α-actin antibody (DSHB, University of Iowa, USA) and HRP-conjugated secondary antibodies (Jackson ImmunoResearch). Enhanced chemiluminescence signals were detected by the ChemiDoc imaging system (Bio-Rad).

For immunostaining, stage 12-16 embryos were devitellinized in 0.1× MMR and fixed in cold Dent's fixative either overnight at 4°C or for 3 h at RT. Fixative was removed by sequential washes at RT in 4 ml of 75%, 50%, and 20% methanol in PBST (PBS+0.1% Triton X-100), each for 5 min. Embryos were then washed three times in 4 ml of 1× PBST at RT (5 min per wash). Blocking was performed in 200 μL of 20% fetal bovine serum (FBS) in 1× PBST for 1 h at RT. The blocking solution was removed, and embryos were incubated overnight at 4°C in 100 μl of primary antibody diluted 1:100 in 10% FBS in 1× PBST. Primary antibodies used were rabbit anti-ADIP, mouse anti-Wtip [DA2B11, Developmental studies hybridoma bank (DSHB)] and mouse anti-ZO1-FITC (Invitrogen) to label cell membranes, rabbit anti-β-catenin antibody C2206 (Sigma-Aldrich), mouse anti-HA antibody (12CA5) and rabbit anti-GFP antibody A6455 (Invitrogen) to detect the HA-myrBFP or H2B-eGFP tracer in MO-injected cells. Following primary antibody incubation, embryos were washed five times in 1xPBST at room temperature (RT), with 1 h intervals between washes. Embryos were then incubated overnight at 4°C with secondary antibodies diluted in 10% FBS in 1xPBST: Alexa Fluor 488–conjugated goat anti-mouse (1:300; Thermo Fisher Scientific) and Cy3-conjugated donkey anti-rabbit (1:400; Jackson ImmunoResearch). Immunostained embryos were washed three times in PBST for 10 min each.

Imaging was performed using BC43 spinning disk confocal microscope (Andor, Oxford Instruments) with a 20x objective lens and Fusion Ver. 2 image processing software. Each experiment was repeated at least three times, with a minimum of ten embryos per experimental group. More than 90% of embryos displayed the described phenotypes in all experimental groups. Z-stack images were processed using the maximum projection function in ImageJ for subsequent analysis and quantification.

### Quantification and statistical analysis

Images of immunostained *Xenopus* superficial ectoderm cells were segmented using the Cellpose algorithm (Stringer et al., 2021). To analyze endogenous ADIP orientation and polarity, Cellpose-generated cell masks were processed using the updated workflow described (https://github.com/sujaynelson/Endogenous-ADIP-PCP-Quantification.git) on the Github repository. In summary, protein puncta were log-transformed, normalized, and thresholded to generate binary masks. Protein clusters were identified using OpenCV and filtered based on size. To quantify puncta polarization, unit vectors were drawn from the centroid of each segmented cell to the centroid of individual protein clusters. These vectors were averaged per cell and weighted by cluster size or fluorescence intensity to compute a polarity vector. The orientation of each polarity vector was then compared to a reference force vector directed toward neighboring constricting cells. Planar polarity was visualized using rose plots, which were generated by binning the angular orientations of polarity vectors into circular histograms, weighted by cluster size or intensity. Statistical significance of angular distributions was assessed using the Chi-square test.

### Acknowledgements

We thank Sujay Nelson for advice on Python scripts for image analysis and quantification and Sokol laboratory members for valuable discussions. We also acknowledge Cocalico help with animal immunization and sera preparation.

### Competing interests

The authors declare no competing or financial interests.

### Author contributions

Conceptualization: S.S.V., S.Y.S.; Data curation: S.S.V.; Formal analysis: S.Y.S.; Funding acquisition: S.Y.S.; Investigation: S.S.V., K.I., C.-W.C., D.A.; Methodology: S.S.V.; Project administration: S.Y.S.; Software: S.S.V.; Supervision: S.Y.S.; Validation: S.S.V., K.I., C.-W.C., D.A.; Visualization: S.S.V.; Writing – original draft: S.S.V., S.Y.S.; Writing – review & editing: K.I., C.-W.C., D.A.

### Funding

This research was supported by the National Institute of General Medical Sciences grant MIRA R35GM122492 to S.Y.S. and the National Institutes of Health grant R24OD038109 to D.A. Open Access funding provided by Icahn School of Medicine at Mount Sinai. Deposited in PMC for immediate release.

## Data and resource availability

The monoclonal antibody DA2B11 has been deposited in Developmental studies hybridoma bank (DSHB). All relevant data and images have been deposited to the Mendeley Data, https://data.mendeley.com/datasets/wz28m3whm4/1 and are publicly available.

## Peer review history

The peer review history is available online at https://journals.biologists.com/bio/lookup/doi/10.1242/bio.062452.reviewer-comments.pdf

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
