## [Peer Review File · Biology Open]

Planar polarization of endogenous ADIP during *Xenopus* neurulation

Satheerja Velayudhan, Keiji Itoh, Chih-Wen Chu, Dominique Alfandari and Sergei Sokol

DOI: 10.1242/bio.062452

Editor: Tristan A Rodriguez

Review timeline

Original submission: 23 December 2025

Editorial decision: 12 January 2026

First revision received: 15 January 2026

Accepted: 16 January 2026

Original submission

First decision letter

MS ID#: bio.062452

MS Title: Planar polarization of endogenous ADIP during *Xenopus* neurulation

Authors: Satheerja Velayudhan, Keiji Itoh, Chih-Wen Chu, Dominique Alfandari and Sergei Sokol

I have now reached a decision on the above manuscript.

The reviewer reports are shown at the bottom of this email.

As you will see, the reviewers gave favourable reports, but raised some critical points that will require amendments to your manuscript. I hope that you will be able to carry these out, because we would like to be able to accept your paper.

At this stage, we also ask you to ensure your manuscript complies with our formatting guidelines - please see our manuscript preparation guidelines for details. Provided you are able to fully address the referees' comments, we are positive about publication of your paper (we accept over 95% of revision submissions) and therefore hope you won't mind any extra work involved in reformatting your manuscript at this point.

Please upload both a 'clean' version of your Word file, along with a highlighted version clearly showing where you have made changes in the revised manuscript. Please avoid using 'Track changes' in Word files as these are lost in PDF conversion.

I should be grateful if you would also provide a point-by-point response detailing how you have dealt with the points raised by the reviewers in the 'Response to Reviewers' box. Please attend to all of the reviewers' comments. If you do not agree with any of their criticisms or suggestions please explain clearly why this is so.

Reviewer 1

Comments for the author

This is a thorough paper. The authors demonstrate that endogenous ADIP forms puncta that polarise during wound repair and neural plate folding, and that this localisation requires the PCP component Diversin, along with intact microtubules and actomyosin networks. Antibody validation and

morpholino knockdown controls are executed well, and the quantitative image-analysis pipeline is appropriate and clearly described. One issue would be the integrity of actin after methanol fixation - could the authors comment on this?

It might be useful to state explicitly the number of replicates or embryos performed.

An expansion of the discussion touching on the context-dependent functions of ADIP and how these may be regulated.

That said this is a nice manuscript and the experiments are done well.

Reviewer 2

Comments for the author

This is a very clearly written and well-illustrated manuscript in which the authors extend their previous findings obtained using ADIP overexpression in *Xenopus* embryos by analysing the distribution of endogenous ADIP protein and combining this with morpholino (MO) knockdown experiments. The authors show that ADIP protein is polarised not only during wound healing but also within the anterior neural plate, where it is enriched toward the midline, a region exposed to higher mechanical tension. Finally, they demonstrate that PCP and cytoskeletal proteins are required for normal ADIP polarisation. To generate these data, the authors have produced new antibodies against ADIP and Wtip, the latter of which has been deposited in the DSHB, representing a valuable resource for the community.

Overall, the data support the authors' conclusions. The images are of high quality and are complemented by appropriate quantitative analyses.

I have only a few minor comments:

1. In the Results section, the references to Figure 1 need to be corrected. While the Figure 1 legend is correct, the in-text citations are not (e.g. Fig. 1D is referred to as Fig. 1C, with similar mismatches continuing through Fig. 1G, which is not mentioned in the text).
2. It would be helpful to briefly describe how the wounds were generated in the embryos.
3. In Figure 2 (panels A-C and D-E), arrows are used but their meaning is not fully explained. Please include a description of what the arrows indicate for each panel.
4. In Figure 3 (line 39), should "anterior border of the neural plate in J-K" be used instead?
5. The number of embryos analysed should be indicated, either in the figure legends or in the main text, with clarification of how many showed the described phenotypes.

Reviewer's Responses to Questions

Experimental quality

Does each figure have the proper controls?

If 'No', please indicate reasons in Comments for Author box below.

Reviewer #1:

- Yes

Reviewer #2:

- Yes

Were the data analyzed using appropriate statistical tests?

If 'No', please indicate reasons in Comments for Author box below.

Reviewer #1:

- Yes

Reviewer #2:

- Yes

Reproducibility

Were experiments performed using adequate number of biological replicates?

If 'No', please indicate reasons in Comments for Author box below.

Reviewer #1:

- Yes

Reviewer #2:

- Yes

Does the methods section provide sufficient detail to permit reproducibility?

If 'No', please indicate reasons in Comments for Author box below.

Reviewer #1:

- Yes

Reviewer #2:

- Yes

Completeness

Are the manuscript's conclusions supported by the data?

If 'No', please indicate reasons in Comments for Author box below.

Reviewer #1:

- Yes

Reviewer #2:

- Yes

Scholarship

Do the authors cite and discuss the merits of data that would argue for and against their conclusion?

If 'No', please indicate reasons in Comments for Author box below.

Reviewer #1:

- Yes

Reviewer #2:

- Yes

Does the manuscript title & abstract accurately reflect the contents of the manuscript, without hyperbole?

If 'No', please indicate reasons in Comments for Author box below.

Reviewer #1:

- Yes

Reviewer #2:

- Yes

First revision

Author response to reviewers' comments

Reviewer 1: This is a thorough paper. The authors demonstrate that endogenous ADIP forms puncta that polarise during wound repair and neural plate folding, and that this localisation requires the PCP component Diversin, along with intact microtubules and actomyosin networks. Antibody validation and morpholino knockdown controls are executed well, and the quantitative image-analysis pipeline is appropriate and clearly described.

1) One issue would be the integrity of actin after methanol fixation - could the authors comment on this?

Methanol fixation is known to extract lipids and disrupt actin microfilaments but it did not alter the localization of the exogenous ADIP when compared in fixed and live samples (see Chu et al. 2025). We also observed a similar ADIP localization after formaldehyde fixation, but it had higher background levels. In contrast, pharmacological inhibition of actomyosin networks in vivo using Cytochalasin D, Blebbistatin, or Y-27632 resulted in a clear loss of ADIP polarity.

2) It might be useful to state explicitly the number of replicates or embryos performed. For each experiment, a minimum of 10 embryos were analyzed, and each experiment was repeated at least three times. In all groups, more than 90% embryos displayed the described phenotypes. This information has been included in methods and figure legends.

3) An expansion of the discussion touching on the context-dependent functions of ADIP and how these may be regulated.

Discussion has been expanded as requested by the reviewer. We point out that “the interaction of ADIP with both microtubular and actomyosin networks suggests that ADIP may function in different tissues in a context-dependent manner. In some epithelial cells, the association of ADIP with known mechanosensitive modulators of cell junctions, α -actinin and Afadin (Asada et al., 2003) may form a network sensing the direction of actomyosin contractions and establishing PCP to coordinate force transmission across tissue. In other tissues, such as mucociliary epithelium, ADIP complexes may be involved in the control of rotational planar polarity of the basal bodies”.

Reviewer 2: This is a very clearly written and well-illustrated manuscript in which the authors extend their previous findings obtained using ADIP overexpression in *Xenopus* embryos by analysing the distribution of endogenous ADIP protein and combining this with morpholino (MO) knockdown experiments. The authors show that ADIP protein is polarised not only during wound healing but also within the anterior neural plate, where it is enriched toward the midline, a region exposed to higher mechanical tension. Finally, they demonstrate that PCP and cytoskeletal proteins are required for normal ADIP polarisation. To generate these data, the authors have produced new antibodies against ADIP and Wtip, the latter of which has been deposited in the DSHB, representing a valuable resource for the community.

Overall, the data support the authors' conclusions. The images are of high quality and are complemented by appropriate quantitative analyses. I have only a few minor comments:

1. In the Results section, the references to Figure 1 need to be corrected. While the Figure 1 legend is correct, the in-text citations are not (e.g. Fig. 1D is referred to as Fig. 1C, with similar mismatches continuing through Fig. 1G, which is not mentioned in the text).

Thank you for pointing out these inconsistencies. We have corrected the in-text citations to Figure 1 as follows: Fig. 1C-C' was changed to Fig. 1D-D', Fig. 1D-D' to Fig. 1E-E', Fig. 1E to Fig. 1F, and Fig. 1F to Fig. 1G. The current references point to the appropriate panels in Figure 1.

2. It would be helpful to briefly describe how the wounds were generated in the embryos.

The following description was added to the Methods section. “Wounds were generated by creating a small circular lesion in the animal epithelium of devitellinized Stage 12 embryos using fine dissection forceps. Embryos were then allowed to heal for 30 minutes in 0.7 \times MMR solution. Following wound healing, embryos were fixed in Dent's fixative and processed for immunostaining to analyze endogenous ADIP polarization.”

3. In Figure 2 (panels A-C and D-E), arrows are used but their meaning is not fully explained. Please include a description of what the arrows indicate for each panel.

Prompted by the reviewer, we now describe arrow annotations in Fig. 2 legend. In Fig. 2A, arrows denote partially colocalized puncta of endogenous ADIP and Wtip. In Fig. 2D, arrows indicate the polarized enrichment of ADIP puncta and the distribution of Wtip at cell junctions facing the wound. In Figs 2B and 2E, arrows represent individual quantified polarity vectors.

4. In Figure 3 (line 39), should "anterior border of the neural plate in J-K" be used instead?

Thank you for noting this. We have corrected the text to read "anterior border of the neural plate in J, K" instead of "J" in Figure 3 legend.

5. The number of embryos analysed should be indicated, either in the figure legends or in the main text, with clarification of how many showed the described phenotypes.

We have now included this information in the revised text. See also our response to Reviewer 1, point 2.

Second decision letter

MS ID#: bio.062452R1

MS Title: Planar polarization of endogenous ADIP during *Xenopus* neurulation

Authors: Satheeja Velayudhan, Keiji Itoh, Chih-Wen Chu, Dominique Alfandari and Sergei Sokol

I am happy to tell you that your manuscript has been accepted for publication in *Biology Open*, pending our standard publication integrity checks. It was accepted on 16th January 2026.